

# Smearing of causality by compositeness divides dispersive approaches into exact ones and precision-limited ones

**Felipe J. Llanes-Estrada⋆ and Raúl Roldán-González**

Dept. Física Teórica & IPARCOS, Universidad Complutense, Madrid, 28040, Spain

⋆ fllanes@fis.ucm.es

## Abstract

Scattering off the edge of a composite particle or finite–range interaction can precede that off its center. An effective theory treatment with pointlike particles and contact interactions must find that the scattered experimental wave is slightly advanced, in violation of causality (the fundamental underlying theory being causal). In practice, partial–wave or other projections of multivariate amplitudes exponentially grow with $\mathrm{Im}(E)$, so that analyticity is not sufficient to obtain a dispersion relation for them, but only for a slightly modified function (the modified relations additionally connect different $J$). This can limit the precision of certain dispersive approaches to compositeness based on Cauchy's theorem. Awareness of this may be of interest to some dispersive tests of the Standard Model with hadrons, and to unitarization methods used to extend electroweak effective theories. Interestingly, the Inverse Amplitude Method is safe (as the inverse amplitude has the opposite, convergent behavior allowing contour closure). Generically, one-dimensional sum rules such as for the photon vacuum polarization, form factors or the Adler function are not affected by this uncertainty; nor are fixed-$t$ dispersion relations, cleverly constructed to avoid it and whose consequences are solid.

# 1   Introduction

Dispersive methods are widely used in nuclear and particle scattering [1] and essential to constrain physics beyond the Standard Model [2]. Often due to the nonperturbativity of strong interactions and the difficulty in calculating therewith, or to ignorance of any underlying theory extending the electroweak Standard Model, amplitudes may not always be tractable from first principles for all energies. Dispersive approaches then allow to constrain the amplitudes with all the information known *ab initio* without access to the underlying Lagrangian dynamics. These constraints are powerful but by no means lead to unique amplitudes. External information is necessary to gain complete amplitudes (whether experimental data, knowledge of subtraction constants from an Effective Lagrangian, or of asymptotic high–energy behavior from other considerations, such as in $\pi\pi$ scattering [3–5].)

   We should distinguish two types of dispersive approaches, with the divide being basically the dimensionality of the problem. The first type typically includes integral representations for functions or correlators of one Lorentz invariant variable *s*, often used as sum rules [6]. These provide crucial tests of many aspects of the Standard Model involving the strong interactions. Starting points in their derivation are usually unitarity and completeness (section 3) such as the use of the optical theorem for the amplitude's imaginary part for physical energies in terms of both elastic and inelastic cross–sections,

$$Im\{\mathcal{M}(i \to X \to i)\} = 2E_{CM}|p_{CM}|\sum_X \sigma(i \to X) \,. \tag{1}$$

(CM refers to the center of mass throughout.) These principles are enough to obtain spectral representations for the functions of interest. Additionally, one can use causality in the form of analiticity in the complex *s* plane, to relate such functions in different processes.

   The second type (section 2), in which we concentrate, involves multivariate functions (thus, more than one propagating particle is involved) and has a stronger focus on causality through Cauchy's theorem, an identity for analytic functions in a complex plane domain:

$$t_J(E) = \frac{1}{2\pi i}\oint_C dE' \frac{t_J(E')}{E' - E} \quad E \in \mathbb{C} \,, \tag{2}$$

here exemplified for a partial-wave scattering amplitude as function of the energy *E* (with identical expressions for $t(E, \theta)$ or other scattering amplitudes). Alternatively to partial waves, one can think of fixed-angle scattering, fixed-*t* scattering, and multiparticle scattering. The description of many processes (such as Compton scattering, Deeply Virtual Compton Scattering, meson production, meson-meson scattering, and also extensions of the electroweak standard model in $W_L - W_L$ scattering) is often reinforced by the use of this type of dispersion relations. The needed analyticity in *E* follows from causality along a well–known line of thought [7], here simplified. The scattering amplitude as a function of energy is the Fourier transform of that which is function of time $\tau$,

$$t_J(E) = \int_{-\infty}^{\infty} \hat{t}_J(\tau)e^{iE\tau}d\tau \,. \tag{3}$$

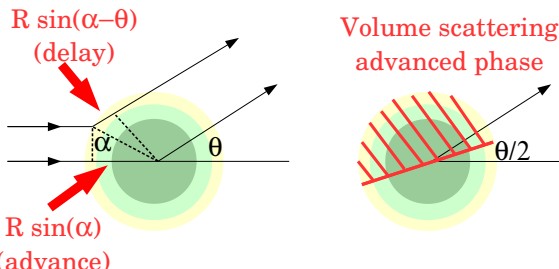

Figure 1: Left: the phase advance of a ray scattered from $r = R$ over one scattered from the center is $R(\sin\alpha - \sin(\alpha - \theta))$. Right: scattering from anywhere in the striped half sphere leads scattering from the center of the circumference and displays apparent violation of causality (the same holds at each plane parallel to the one depicted, only with diminished $R$).

If the incoming wavepacket hits a pointlike target at $\tau = 0$, causality entails that $\hat{t}_J(\tau) = 0$ for $\tau < 0$. Therefore, as the integrand vanishes for earlier times, the lower integration limit can be set to 0. Extension of $t_J(E)$ to the complex plane allows to write

$$t_J(E) = \int_0^\infty \hat{t}_J(\tau) e^{i\mathcal{R}e\{E\}\tau} e^{-\mathcal{I}m\{E\}\tau} d\tau \ . \tag{4}$$

The last exponential ensures convergence in the upper half–$E$ complex plane, and an analytic $t_J(E)$ (Titchmarsh's theorem makes the statement rigorous) that is well behaved for $\mathcal{I}m\{E\} \to +\infty$, allowing use of Cauchy's theorem by closing an infinite semicircular contour.

In section 2 we discuss the resulting dispersion relation and an example numeric evaluation of the uncertainty introduced by slightly relaxing causality for $t_J(\tau)$ nonvanishing at times a bit earlier than $\tau = 0$. But first, in subsection 1.1 we recall the basic discussion [8, 9]; a more rigorous treatment of the underlying theory can be found in Nussenzveig's book [10]. Because the numeric consequences of this apparent violation of causality are not computable in a straightforward manner, as they depend on target structure and underlying interaction, our goal is limited to unveiling it as an uncertainty in the resulting dispersion relations for multivariate scattering amplitudes.

## 1.1 Advanced scattering for composite objects

For simplicity, take a beam of pointlike objects (photons serve as example) scattering an angle $\theta$, with $x = \cos\theta$, off a composite target [1] as depicted in fig. 1.

The scattering can happen at a distance $R$ from the target's center of mass, at a point with visual therefrom forming an angle $\beta \equiv \alpha + \pi/2$ with the direction of incidence. The target softness and underlying interaction details determine the probability of such scattering configuration, $P(R, \alpha; \theta)$. In the usual asymptotic analysis, $R$ and $\alpha$ are implicitly integrated over and only the dependence with $\theta$ remains; this carries over to the Effective Theory where $R = 0$. Nevertheless, at order $R$, we have an apparent violation of causality because the scattering off $R$ can appear at $\tau = +\infty$ with a phase ahead of the scattering from the center. As shown in its left plot (limited to plane geometry, since planes parallel to that in the figure only differ in a decreased $R$), off–center scattering advances the phase due to the path difference

$$R\ (\sin\alpha - \sin(\alpha - \theta)) = 2R\sin\left(\frac{\theta}{2}\right)\cos\left(\alpha - \frac{\theta}{2}\right). \tag{5}$$

---

[1]The target should not be thought of as a rigid ball: it is enough that any surface at distance $R$ from its center scatters the projectile towards an angle $\theta \neq 0$. This is certainly the case for hadrons.

The advanced wave could have scattered from any point with angle to the visual $\beta \in (\theta/2, \pi + \theta/2)$; its $2R\sin(\theta/2)$ maximum occurs in the middle of that $\beta$ interval. Because of this path difference the scattered amplitude does not vanish for $\tau < 0$; $\hat{t}_J(\tau) = 0$ is only guaranteed for $\tau < -\frac{2R}{c}\sin\frac{\theta}{2}$. (Subsequently, $c = 1$ is set.) Of course, this inequality is smeared by the target's softness so that $R$ is distributed, but to discuss *uncertainties* in $R = 0$ computations it is sufficient to use a typical $R$.

That the scattering amplitude $t_J$ or similar is still analytic can be read off Eq. (4). The lower integration limit then needs to be set to $-2\frac{R}{c}\sin(\theta/2) < 0$ since the time-dependent amplitude only vanishes for earlier times. This change does not affect the convergence at $\tau \to \infty$, so that the resulting $t_J(E)$ is still analytic in the upper-energy plane; but it grows (exponentially) for increasingly positive $Im(E)$. That is, a finite target yields an analytic function but a non-convergent contour integral. Only when the signal can be arbitrarily advanced, and $\tau \to -\infty$ needs to be taken in the time integration, is analyticity completely lost.

This advanced-wave phenomenon for composite objects requires multidimensional geometry. In one dimension, although an interaction may be triggered upstream of the center of mass, the outgoing signal still has to go through that center of mass in its propagation, so it will not get ahead of the wave nominally scattered at the center of the system. This is why dispersion relations based on forward scattering, or for intrinsically one-dimensional problems such as propagators or current-current correlators $i \int d^4x e^{iq\cdot x}\langle 0|T(J(x)J(0)^\dagger)|0\rangle$ are unaffected.

On the other hand, any scattering process in 2 or greater dimension, where one or both objects are of finite size, or where the interaction is finite range, will suffer from that exponential behaviour in the complex plane and care with the formulation of dispersion relations will be needed. Examples include: photon-hadron scattering, hadron-hadron scattering, photon-atom scattering, nuclear scattering, etc.

## 2 Causality–driven dispersion with more than one variable: $\pi\pi$, (or $W_L W_L$) elastic scattering

We examine quasiGoldstone-boson scattering as an example of a dispersion relation eventually taking microscopic–physics dependent corrections, and amenable to clear Effective Field Theory treatment. We have in mind two possible physical systems: $\pi\pi$ scattering, interesting because of the much extant data and many existing analysis, and of importance at the precision frontier, and longitudinal W/Z scattering, of interest for searches of new physics at the energy frontier. When we speak of "Effective Theory" we have either of the two relevant theories for these physical systems, Chiral Perturbation Theory [11] or Higgs Effective Field Theory [13] that hide in local fields (in a multipole-like expansion) any compositeness of underlying renormalizable theories.

The kinematic variables are Mandelstam's invariants $s$, $u$, and $t = -(1-x)(s-4m^2)/2$. Since $s = E_{CM}^2$, its extension to the complex plane sees its phase linked to that of the energy by $\theta_s = 2\theta_E$.

Because of Eq. (4), the amplitude is analytic for $\mathcal{I}mE > 0$ or $\theta_E \in (0, \pi)$, and thus, in the entire complex $s$-plane except for cuts (and eventually poles, though not for $\pi\pi$ scattering) on the real axis. With $\tau > 0$, the factor $e^{-\mathcal{I}m\{E\}\tau}$ damps the amplitude over the large semicircle $\mathcal{I}m\{E\} \gg 0$ and therefore over the entire circle $|\mathcal{I}m\{s\}| \gg 0$.

Cauchy's theorem becomes the integral fixed-$t$ dispersion relation

$$T(s,t,u) = \frac{1}{\pi}\left(\int_{4m^2}^{\infty} + \int_{-\infty}^{t<0}\right)ds'\frac{Im\{T(s',t,u)\}}{s'-s-i\epsilon}, \tag{6}$$

that can be subtracted as needed and allows to proceed from the amplitude over the physical

$s > 4m^2$ (right cut) and unphysical $s < 0$ (left cut) to complex $s$. There are similar relations for amplitudes at fixed scattering angle $T(E, \theta)$ and for the partial–wave projected amplitudes $t_J$: this last one, with $n$ subtractions, is the well known

$$t_J(s) = \sum_{k=0}^{n-1} \frac{t_J^{(k)}(0)}{k!} s^k + \frac{s^n}{\pi} \left( \int_{4m^2}^{\infty} + \int_{-\infty}^{0} \right) \frac{dz}{z^n} \frac{Im\{t_J(z)\}}{z - s - i\epsilon} \tag{7}$$

valid for point particles with contact interactions. But if the interaction occurs with the CM of the two pions separated a finite range $R$, from Eq. (4) with a lower limit that is not 0 but the advanced time of subsec. 1.1, the amplitude may pick up a phase-advance contribution proportional to $e^{-2iR\sqrt{s-4m^2}\sin(\theta(s,t)/2)}$ for each $R$ layer scattering ahead of the CM in the direction of $\theta$.

At large $\mathcal{I}mE > 0$ such factors diverge: the standard application of Cauchy's theorem with a circular contour at infinity is in question because the integral over the semicircle at infinity can then also diverge. An exception does of course happen when the exponential contribution carrying $R$ is suppressed by the $\sin(\theta)$ vanishing at forward angles: there, Regge kinematics showing a power-law, and not a rotating phase, dominates hadron scattering (and experimental data shows a smooth cross section). The possible unchecked-exponential amplitude growth requires an imaginary part of $s$, so it is not easily seen in the data at real $s$ (it could perhaps be tested with an appropriately constructed sum rule, but this exceeds our present effort). In any case, forward dispersion relations do not suffer an uncertainty since they are formulated for $\theta = 0$ and the exponential factor becomes just a troubleless unit factor. For other angles, or for the partial waves, the standard dispersion relation is not guaranteed.

The exponential, with $\sin(\theta/2) = \sqrt{(1-x)/2}$, is an irrelevance [10] for fixed $t$ since it becomes a fixed constant $exp(2iR\sqrt{|t|})$ so that Eq. (6) is still valid. In addition to forward dispersion relations, fixed-$t$ dispersion relations are adequate with composite objects, which is a classic result.

But upon proceeding to a fixed reference frame and fixing the angle [2] (except for forward ($\theta = 0 = t$) dispersion relations) or, for the partial waves, its conjugate variable $J$, modification is required.

## 2.1 Use of auxiliary functions to obtain information on the physical amplitude

Standard use of Cauchy's theorem requires a function with good behavior for $\mathcal{I}mE \gg 0$. One possibility is to introduce an auxiliary function for which the diverging exponential behaviour is not present. Then the dispersion relation can be written for it, and afterwards one can try to extract information on the physical amplitude from such relation. One auxiliary function that can be chosen (*not* a scattering amplitude) is the modified

$$T(s, t, u) \rightarrow e^{2iR\sqrt{s-4m^2}\sin(\theta(s,t)/2)} T(s, t, u) . \tag{8}$$

The square root in Eq. (8) adds to the right discontinuity in the resulting dispersion relation which replaces Eq. (7). At finite $R$, the auxiliary partial wave projections are

$$t_J'(s) = \int_{-1}^{1} dx \frac{P_J(x)}{64\pi} e^{2iR\sqrt{s-4m^2}\sqrt{\frac{1-x}{2}}} T(s, t(x), u(x)) \tag{9}$$

where, for a moment, we only keep one order in $R$

$$e^{2iR\sqrt{z-4m^2}\sqrt{\frac{1-x}{2}}} \simeq 1 + 2iR\sqrt{z-4m^2}\sqrt{\frac{1-x}{2}} . \tag{10}$$

---

[2]This was observed early on by Gell-Mann, Goldberger and Thirring, see discussion around Eq.(4.15) of their 1954 work [15].

The Left Hand Side of Eq. (7) then takes a correction $t_J(s) \rightarrow t_J(s) + \Delta t'_J(s)$ (with poles taken as $\frac{1}{z-(s+i\epsilon)}$; minimum additional discussion on Schwarz's reflection principle is deferred to appendix A):

$$
\begin{aligned}
\Delta t'_J(s) = {} & 2R \frac{s^n}{\pi} \left[ \int_{4m^2}^{\infty} dz \frac{\sqrt{z-4m^2}}{z^n(z-s)} \sum_{L=0}^{\infty} A_{JL} Re\{t_L(z)\} + \int_{-\infty}^{0} dz \frac{\sqrt{z-4m^2}}{z^n(z-s)} \sum_{L=0}^{\infty} A_{JL} Im\{t_L(z)\} \right] \\
& + \sum_{k=0}^{n-1} \tilde{t}_J^{(k)}(0) \frac{s^k}{k!} ,
\end{aligned}
\tag{11}
$$

which is actually dependent on partial waves of different angular momentum through the (asymmetric) matrix

$$
A_{JL} = \frac{2L+1}{2} \int_{-1}^{1} dx P_J(x) P_L(x) \sqrt{\frac{1-x}{2}} . \tag{12}
$$

Since $\sqrt{\frac{1-x}{2}}$ is of slow variation, one expects that very different $J$ and $L$ are weakly coupled by the cancellations among Legendre polynomials. The diagonal $A_{J=L}$ elements are between 0.6 and $\frac{2}{3}$ while the off-diagonal ones fall rather quickly with $J-L$, for example, $A_{02} \simeq -0.095$. In turn, the subtraction constants in Eq. (11) are $\tilde{t}_J^{(k)} = t_J^{(k)'} - t_J^{(k)}$ and carry $R$–dependence. When ignoring $R$ and employing dispersion relations with data fits, the $R = 0$ subtraction constants are probably absorbing part of the total uncertainty, so we can use what is left of them, the $\tilde{t}_J^{(k)}$, to minimize it.

That the partial waves are mixed is a phenomenon seen before, in the context of the Roy or the Baacke-Steiner equations [16, 17]. The difference is that the Roy equations are obtained to eliminate the left cut in terms of the right cut of crossed channels, employing crossing; they relate the amplitude in several isospin channels, additionally to several angular momenta, so they are not strictly speaking dispersion relations, though they feature integrals along the right cut. In Eq. (11) even with the left cut untouched, the partial waves are mixed for the physical scattering amplitude (because the auxiliary function that satisfies a dispersion relation exactly differs by an angle-dependent exponential) [3].

## 2.2 A worked numerical example in pion scattering

Let us show the typical size of the uncertainty induced by Eq. (11) in the $\pi\pi$ case if a plain dispersion relation for $t_J$ is invoked without support from fixed-$t$ dispersion relations: for this we limit ourselves to the right–cut integral from $4m^2$ on, where the scalar amplitude is well known [3–5]. We plot its real part, with characteristic dragon shape, in figure 2. For a quick estimate we adopt as effective range of the interaction $R \simeq m_\sigma^{-1} \simeq 2$ GeV$^{-1}$ (compare with 0.79 fm $\simeq 4$ GeV$^{-1}$ for the pion scalar radius [19, 20] appropriate for $J = 0$ or with $1/m_\rho = 0.26$ fm for the vector one, $J = 1$). The first two terms in the expansion are not representative of the exponential in Eq. (10) at energies much beyond threshold, so we limit ourselves to that area. The outcome is plot in figure 3. We have chosen $n = 1$ and used this one subtraction to make the uncertainty vanish at threshold. However, the uncertainty band quickly grows with $E$.

---

[3]It is worth remarking here that the Roy and Steiner equations are derived from fixed-$t$ dispersion relations, so they are not subject to corrections either, as long as the conditions for the convergence of the partial wave series are satisfied [18].

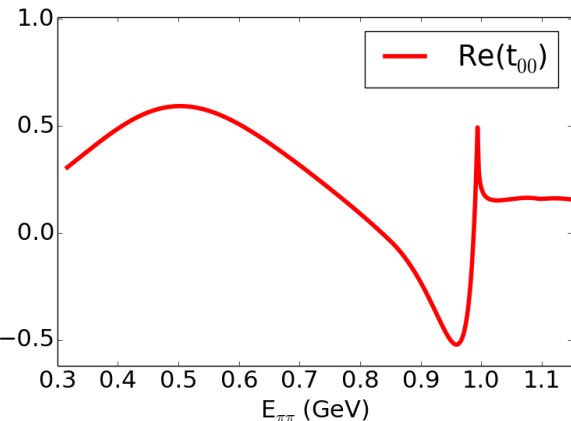

Figure 2: Real part of the $\pi\pi\ t_0$ partial–wave amplitude [3–5] employed to compute the right cut in Eq. (11).

Therefore, we proceed to reanalyzing the full exponential. We then find, up to $J = 2$ (the effect of the $d$–wave is small, but we include it nevertheless),

$$
\Delta t_0'(s) = \sum_{k=0}^{n-1} \tilde{t}_J^{(k)}(0)\frac{s^k}{k!} + iIm\left[t_0(F_0(s)-1)+t_2F_2(s)\right]
$$
$$
+ \frac{s^n}{\pi}\left(\text{PV}\int_{4m^2}^{\infty} + \int_{-\infty}^{0}\right)\frac{dz}{z^n}Im\left(t_0(F_0(z)-1)+t_2F_2(z)\right),
$$

(13)

with $F_J(s) \equiv \frac{(2J+1)}{2}\int_{-1}^{1}dx\,e^{2iR\sqrt{s-4m^2}\sqrt{(1-x)/2}}P_J(x)$, and $PV$ the principal value integral. An example numerical computation of Eq. (13), twice subtracted, is seen in fig. 4. Once more, the uncertainty induced is not negligible, because $R$ is quite large (the compositeness scale, $R^{-1}$, is comparable to the scattering energies).

These considerations have only provided the difference between the right hand cut of a standard partial wave dispersion relation and an $R$-modified one; it is far from our intention to attempt an equivalent computation of the left hand cut, that is notoriously difficult; only known with some confidence in the nonrelativistic approximation [21]; and whose contribution in the resonance region of energy of interest for the LHC, deep in the right hand cut, is suppressed anyway by the structure of the dispersion relation. Given the uncertainty in the left hand cut, that it is eventually constrained by crossing from a set of different reactions and partial waves, it is reasonable to assume that the uncertainty induced by it will add up linearly to that of the right-hand cut

$$
\Delta t(s) = |\Delta_{\text{LC}}t(s)| + |\Delta_{\text{RC}}t(s)|,
$$

so that the uncertainty induced by the right cut, already sizeable, is a lower bound on the total uncertainty.

These plots do not mean that a parametrization of experimental data needs to be so uncertain; they should be interpreted as the uncertainty when using a partial-wave dispersion relation to describe data, absent an underlying fixed-$t$ dispersion relation to shore up the computation.

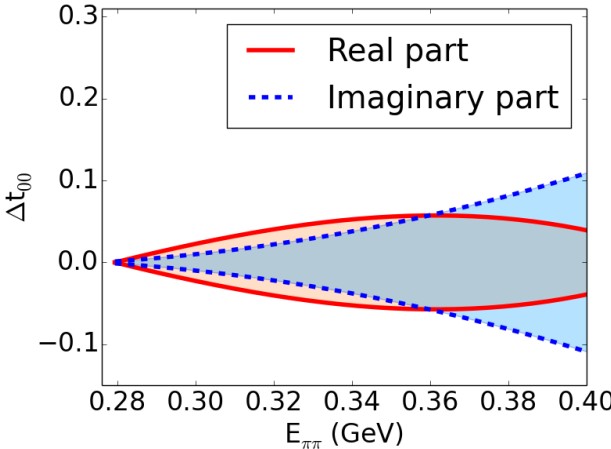

Figure 3: Numerical computation of Eq. (11) (only the right cut with one subtraction, with constant chosen to cancel the effect at threshold), to be understood as a theoretical uncertainty in the real and imaginary parts, respectively, of Eq. (6).

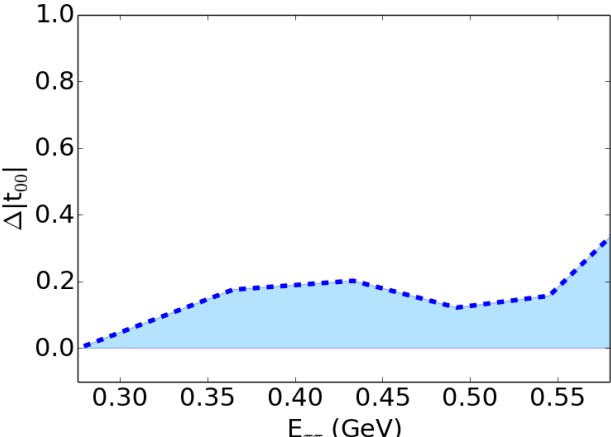

Figure 4: Numerical estimate of Eq. (13) (only the right cut with two subtractions, chosen above threshold); we plot the theoretical uncertainty on the modulus of $t$.

## 2.3 Additional comments on $W_L W_L$ scattering

Future $W_L W_L$ scattering data at the high-energy frontier could be influenced by a composite Higgs sector or other strongly interacting new physics [23]. This could be tested against a fixed-angle or a partial-wave dispersion relation such as Eq.(7): a failure thereof would be indicative of the compositeness of the Goldstone bosons $\omega^i \sim W_L^i$, and reveal a scale $R$ that might otherwise go unnoticed in the absence of a direct resonant manifestation.

More realistically, and as has been the case in hadron physics, the underlying scale would first appear in other, less data-demanding analysis. Hints on the scale of compositeness will appear, if this is how the electroweak sector works, in separations from the Standard Model in EFT couplings following specific patterns. But then, even if that compositeness size times the transfered energy $R \times E$ may be small at an accelerator with insufficient resolution to probe the $W_L$ size, when closing the Cauchy contour in the complex plane, the exponent $R \times \text{Im}(E)$ becomes arbitrarily large. Certain dispersion relations then fail due to the presence of an underlying structure even when this may not yet be probed at available energy.

This scale of compositeness can be unrelated to the range of interaction if the mass of the force carrier is very dissimilar to the mass of the source (e.g. the pion and the nucleon). Our considerations apply to two different scenarios: (a) finite range interactions with pointlike particles, and (b) apparently zero range interactions (local delta functions) when the object is composite at any scale (no matter how small, $\text{Im}E \times R$ will eventually grow beyond order one for large enough imaginary energy).

Finally, the discussion in this section has been strictly kept at the level of $S$-matrix theory to maintain generality. One may wish to see how the exponential factor obstructing the application of Cauchy's theorem in various settings arises within the particular but important case of a quantum field theory. To expose it, a worked example is given in appendix B. But $S$-matrix theory applies in more general scenarios (string theory, nonrelativistic quantum mechanics with fixed particle number, and others).

## 3 One-dimensional spectral representations (such as in the muon's $g-2$)

Spectral dispersive approaches driven by unitarity and completeness that do not require analytic extension far into the complex plane with contours at infinity are not immediately affected by the finite size of the objects under study; neither are essentially univariate problems as advanced at the end of subsection 1.1. In this section we establish that the trouble with large ImE seen in multivariate scattering amplitudes is absent from these scattering-angle free amplitudes.

A case in point that illustrates both observations is the hadron vacuum polarization contribution to the magnetic moment of the muon. The muon's Landé $g$ factor is $\vec{\mu} = g\frac{e\hbar}{2m}\frac{\vec{S}}{\hbar} = g\frac{\mu_B}{\hbar}\vec{S}$, with $\mu_B$ analogous to the Bohr magneton but using the muon's mass $m = 105$ MeV instead. Among other corrections to the Dirac value $g = 2$, those from the strong interactions arise at lowest order from the typical diagram in figure 5.

The $\gamma$ polarization in the diagram includes intermediate $\pi\pi$ states (and more massive hadrons). It appears in the propagator $\triangle'_F(x-y) = \langle 0|T(A(x)A(y))|0\rangle$ (Minkowski indices omitted) with time ordering

$$T(A(x)A(y)) = \theta(x^0 - y^0)A(x)A(y) + \theta(y^0 - x^0)A(y)A(x).\tag{14}$$

The standard treatment [1, 22] proceeds by inserting a complete set of states $\sum_s |s\rangle\langle s| = \mathbb{1}$ with the quantum numbers of the photon field, and exploiting Poincaré invariance to define a spectral density function

$$(2\pi)^{-3}\rho(p^2) \equiv \sum_s \delta(p_s - p)|\langle 0|A(0)|s\rangle|^2.\tag{15}$$

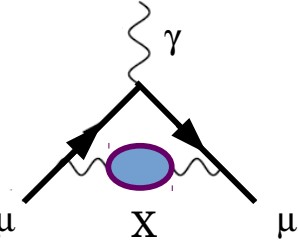

Figure 5: Vertex diagram correcting the muons magnetic moment. $X$ represents the photon vacuum polarization which includes strongly interacting intermediate states.

Extracting the one–photon state by $\rho(p^2) = \delta(p^2) + \sigma(p^2)$, one obtains the propagator's Lehmann representation

$$(2\pi)^4 \, \triangle'_F (k^2) = \frac{-1}{k^2 + i\epsilon} + \int da^2 \frac{\sigma(a^2)}{a^2 - k^2 - i\epsilon} \, , \tag{16}$$

with its typically dispersive form, integrating over a spectral density over the real axis.

To obtain that form [22], the causality condition

$$[A^{\text{free}}(x), A^{\text{free}}(y)] = 0 \ \ \text{if } (\text{x} - \text{y})^2 < 0 \tag{17}$$

has been invoked for the free fields, related to the $i\epsilon$ prescription in the free propagator contained in Eq. (16). Causality appears factorized, satisfied independently of the spectral density (what intermediate states there are and whether they are composite). In fact, the vacuum expectation value of the commutator for the interacting fields is a convolution over $a$

$$\langle 0|[A_\mu(x), A_\nu(y)]|0\rangle = \int_0^\infty da^2 \eta_{\mu\nu} \rho(a^2)$$
$$\times \int \frac{d^4 p}{(2\pi)^3} \theta(p^0) \delta(p^2 - a^2)[e^{-ip(x-y)} - e^{ip(x-y)}] \, ; \tag{18}$$

the factor in the first line carries the spectral density, and the one in the second line enforces causality for any $a$ independently of that density. The propagation of the photon, happening along a straight line, is not altered by any finite radius of intermediate states since forward scattering cannot be advanced by it.

Returning to the muon, the EM vertex coupling $\Gamma^\mu = \gamma^\mu F_1(q^2) + \frac{i\sigma^{\mu\nu}q_\nu}{2m} F_2(q^2)$ leads to $g = 2(F_1(0) + F_2(0)) = 2(1 + F_2(0))$ so that $F_2$ provides the anomalous magnetic moment, and further standard manipulation [24] yields a correction

$$a_\mu = \frac{\alpha}{\pi} \int da^2 \sigma(a^2) \int_0^1 du \left[ \frac{(1-u)u^2}{(1-u)\frac{a^2}{m^2} + u^2} \right] . \tag{19}$$

The spectral density therein provides the vacuum polarization as $\sigma(a^2) = \frac{Im\{\Pi_h\}}{a^2}$ and its hadron contribution can be obtained from a measurable cross–section via the optical theorem (unitarity) $\sigma(e^- e^+ \to h) = \frac{4\pi\alpha}{a^2} Im\{\Pi_h(a^2)\}$, which is the basis of modern analysis of the muon's $g - 2$ [25–27].

In the entire chain of reasoning, which leans on the completeness of the intermediate states and unitarity, there is no room for small apparent violations of causality interfering with the result in Eq. (19). The reason is that Cauchy's theorem has not been employed with a contour over the upper half of the $s$–complex plane where the exponential obstacle requiring modification as in Eq. (8) can appear. This applies to problems in more variables: at the level of a spectral representation, compositeness does not seem to present an obstacle.

As for the use of analyticity, to close a contour in the complex plane is safe in univariate problems. The geometry does not allow the scattering from the object's leading side to be advanced respect to the scattering at the center of mass. Thus, a vacuum polarization function, the Adler function or the pion form factor, all functions of only one variable $s$, can be extended between the timelike and spacelike domains, for example, which leads to important tests of the Standard Model.

To conclude this example, tough other pieces of a complete calculation of the muon's $g - 2$ might be subject to small finite range corrections, as they could involve multiparticle kernels, the cornerstone extraction of its largest hadron contribution seems free from them.

# 4 Conclusion

We have shown how compositeness and more generally noncontact interactions introduce corrections to dispersive approaches based on causality, an observation relevant for the LHC program in which possible deviations from the Standard Model would suggest the use of such dispersion relations to extrapolate to and make predictions about the new physics scale [28], and where compositeness is of interest [29–32].

Such corrections (aspherical, mixing partial waves) vanish in the limit $R \to 0$, see Eq. (11), which is consistent with the literature on Effective Theories. In the limit that the compositeness length vanishes, the resulting EFT is causal [33]. A strict Wigner bound then appears constraining the phase shift $\delta$ to have nonnegative derivative [34]. For a composite object with typical radius $R$, the bound is relaxed to $d\delta/dk > -R$. Nevertheless, this still constrains the effective range expansion [35], though less strongly.

We suggest that this smearing of causality extends to higher energy approaches. Dispersion relations also constrain amplitudes; but for finite $R$, also less strongly so.

This can be the case for approaches that require closing a contour in the complex $s$–plane to apply Cauchy's theorem, because the finite range causes an obstruction. Dispersive approaches in which the integral over the physical cut appears as a consequence of a one-dimensional spectral expansion are not affected by this observation, particularly those addressing the hadron vacuum polarization necessary for the $g - 2$ of the muon.

One of the more widely used dispersive approaches, the Inverse Amplitude Method [36], fairly uses a dispersion relation, since the function for which a contour is closed in the complex $s$–plane is $G \simeq \frac{t_0^2}{t}$ (with $t \simeq t_0 + t_1 + \dots$ being the expansion of the partial wave amplitude in chiral perturbation theory). If the imaginary part of $s$ is large, $G \sim s^2 e^{-2R\sqrt{s}}$ and the great semicircle integral in the Cauchy contour converges.

Likewise, approaches based on fixed–$t$ dispersion relations can be used to obtain a dispersion relation for the partial waves as long as the partial wave expansion itself converges, which is safe in certain kinematic regions.

In any case, even if the dispersion relation underlying a given approach to the amplitude is convergent, one wonders how large would the modification be if, simultaneously, the modified dispersion relation for $t'_J(s)$ defined in Eq. (9), which is certainly valid, is imposed. That is, not only $t_J$ in these safe cases has to satisfy an integral identity, but also the $t'_J$ built from it. In fact, Eq. (8) is the minimal one in the sense that it removes only the strictly needed exponential factor, but it is not unique; one can for example multiply it by a polinomial in $s$, $t$ and $u$ and obtain a whole family of dispersion relations that should be satisfied. This technique of introducing a polynomial for generality is widely used, for example in the context of the Omnès-Mushkelishvili relation, as in [19, 20, 37, 38]. One then needs to make sure that the convergence of the integral on the real axis is adequate, subtracting as necessary. There is ample room for future investigation here.

The catch is that, both in these and the other, more affected dispersion relations, it is not clear to us how our results can be moved from estimates of the introduced theoretical uncertainty, which to our knowledge had never been numerically evaluated, to actual computed corrections that improve predictions. Perhaps one could minimize the separation from the modified dispersion relation using the amplitude parameters, simultaneously with other constraints, but an important problem to solve is the spread in $R$ of the wavepacket's interaction with the target. Further investigation appears necessary.

Perhaps one could construct a family of $R$-dependent dispersion relations, all of which have to be satisfied by the partial wave amplitudes with decreasing level of confidence as $R$ increases, and optimize the fits to minimize the joint deviation from their satisfaction. This might be useful outside the kinematic domain where fixed-$t$ dispersion relations apply.

The reader may wonder how the discussion herein is not widely presented: after all, dispersion relations are supposed to be well established in QCD. A precis can for example be found in the work of Oehme [39, 40] who shows that the dispersion relations known before the advent of gauge theories continue being valid in QCD. His discussion validates the absence of quark and gluon anomalous thresholds due to confinement, but they would not affect the amplitude far from the real axis anyway. Second, he establishes spectral representations due to unitarity and completeness, as we have been discussing in section 3: here the amplitude is expressed in terms of a spectral function, but not in terms of its imaginary part. And finally, Oehme finds dispersion relations à la Kramers-Kronig, relating real and imaginary parts of an amplitude within QCD, but these are forward dispersion relations at zero scattering angle, where the exponential factor that we comment on becomes unity: as we have argued, both fixed-t and forward dispersion relations are unaffected.

What Oehme does not address is dispersion relations that did not work before QCD, so [10] continues being the relevant reference; and, particularly, he does not establish dispersion relations for partial-wave amplitudes. Therefore, his discussion does not impact our observations.

In summary, dispersion relations fall in two classes: those mainly traceable to unitarity and completeness (spectral representations), about which we make no comment; and those derivable from causality by use of Cauchy's theorem and the closing of a large circle in the complex $s$-plane. These are affected if the scatterers are composite, because causality is apparently violated in an effective theory in which their size is discarded, and need more careful examination. Among these, forward- and fixed-$t$-dispersion relations are trouble-free because the phase advance due to the back of the target does not grow with $s$. Other types of dispersion relations, for example those for partial waves, may however carry an $R$-related uncertainty.

## Acknowledgments

We thank Jose R. Pelaez for important feedback and A. Dobado for an early reading of the manuscript.

**Funding information** Work supported by grants MINECO:FPA2016-75654-C2-1-P and MICINN: PID2019-108655GB-I00, PID2019-106080GB-C21 (Spain); Univ. Complutense de Madrid under research group 910309 and IPARCOS institute; the EU's Horizon 2020 programme, grant 824093 STRONG2020; and the VBSCan COST action CA16108.

## A  Schwarz's reflection principle in the derivation of Eq. (11)

Here we briefly discuss the discontinuities of the auxiliary function leading to Eq. (11). There are two modifications to the usual discussion. The first is the appearance of the square root $\sqrt{z - 4m^2}$ coming from the exponent in Eq. (8). This together with the $iR$ factor of the first order Taylor expansion of the exponential exchanges the roles of the real and imaginary parts of the original amplitude $T$.

The integral over a cut for a function with a discontinuity generically takes the form

$$\int_{4m^2}^{\infty} (F_+ - F_-)dz \, ,$$

with the function respectively evaluated on the upper ($+$) and lower ($-$) edges of the cut.

The structure of our auxiliary function $F$ is a product of a square root, whose cut is chosen as usual in this field for positive $s$ or $z$, so that $\arg(\text{sqrt}) \in (0, 2\pi)$, and the scattering amplitude

$T$ that satisfies Schwarz's reflection principle $f(z) = f^*(z^*)$. Let us call these two pieces $f_2$ and $f_1$ respectively, satisfying:

a)  $f_2$ is real along the positive real axis, but it is cut $f_{2-} = -f_{2+}$.

b)  $f_1$ is complex but satisfies Schwarz's reflection principle so that its real part is continuous across the cut and the imaginary part satisfies $\text{Im} f_{1-} = -\text{Im} f_{1+}$.

We can now addres the discontinuity across the right cut,

$$
\begin{aligned}
\text{Disc}(if_1 f_2) &= \text{Disc}(if_2 \text{Re} f_1 - f_2 \text{Im} f_1) \\
&= 2if_2 \text{Re} f_1 \,.
\end{aligned}
\tag{20}
$$

Finally, below the cut of the square root, $f_2$, is continuous, and any discontinuity comes from the amplitude $T$ ($f_1$) alone, so the cut works as needed to establish Eq. (11).

# B  A worked example showing the $R$-dependent exponential

Some readers may find useful to think about the size-dependent phase (that turns into a growing exponential for imaginary energy, possibly obstructing the use of Cauchy's theorem) in terms of a full quantum formalism. One of the simplest examples is the scattering of a scalar particle, with field $\phi_1(x)$, from the bound state of two scalar particles $\phi_2(x)$ and $\phi_3(x)$, all distinguishable for simplicity. The Lagrangian density relevant for that scattering can be taken as

$$
\mathcal{L}_{\text{sc}} = \frac{1}{2} \sum_{i=1}^{3} \left( m_i \phi_i^2 + \partial_\mu \phi_i \partial^\mu \phi_i \right) + \phi_1^2 (g_2 \phi_2^2 + g_3 \phi_3^2) \,.
\tag{21}
$$

This is supplemented by an unspecified nonperturbative, perhaps confining, interaction between $\phi_2$ and $\phi_3$ whose only role is to provide a $2-3$ bound state $|\psi_{23}\rangle$ of two particles. This bound state is taken to have particle 2 in a wavepacket spread around $\mathbf{X}_2 = (0, 0, -R/2)$ and likewise particle 3 around $\mathbf{X}_3 = (0, 0, +R/2) = -\mathbf{X}_2$.

To exemplify, we take two equal Gaussian wavepackets

$$
\langle \mathbf{x}_2 \mathbf{x}_3 | \psi_{23} \rangle = \frac{1}{(\pi^3 a^6)^{1/2}} e^{-\frac{(\mathbf{x}_2 - \mathbf{X}_2)^2 + (\mathbf{x}_3 - \mathbf{X}_3)^2}{2a^2}} \,,
\tag{22}
$$

so that Fourier transforming to momentum space allows us to write the second-quantized state as

$$
|\psi_{23}\rangle = \int \frac{d^3 q_2}{(2\pi)^{3/2}} \frac{d^3 q_3}{(2\pi)^{3/2}} \sqrt{\frac{a^6}{\pi^3}} e^{-\frac{a^2}{2}(\mathbf{q}_2^2 + \mathbf{q}_3^2)} e^{-i\mathbf{q}_2 \cdot \mathbf{X}_2} e^{-i\mathbf{q}_3 \cdot \mathbf{X}_3} a_{q_2}^\dagger b_{q_3}^\dagger |0\rangle \,.
\tag{23}
$$

(Obviously, $a^\dagger$ and $b^\dagger$ are the particle creation operators associated with the fields $\phi_2$ and $\phi_3$, satisfying usual commutation relations, $[a_q, a_k^\dagger] = (2\pi)^3 \delta^{(3)}(\mathbf{q} - \mathbf{k})$ and similarly for $b$). Of note in Eq. (23) is the phase factor $e^{-i\mathbf{q}_2 \cdot \mathbf{X}_2} e^{-i\mathbf{q}_3 \cdot \mathbf{X}_3} = e^{-\frac{iR}{2}(q_3 - q_2)_z}$ that comes from the Fourier transform of the extended object. In an effective theory of small $R$, or multipole expansion $R \to 0$, it could be neglected. But for a finite-sized object it will lead to the exponential factor discussed in section (2).

The model setup for this example scattering process is captured in figure (6).

Denoting the momentum modes of the field $\phi_1$ by $c$ and $c^\dagger$, the part of the Lagrangian in Eq. (21) responsible for the two connected Born scattering diagrams in first order perturbation theory over $g_2 \sim g_3$ is

$$
V := 4 \left( \prod_{i=1}^{4} \int \frac{d^3 k_i}{(2\pi)^3} \right) (2\pi)^3 \delta^{(3)}(\mathbf{k}_2 - \mathbf{k}_1 + \mathbf{k}_4 - \mathbf{k}_3) c_{k_1}^\dagger c_{k_2} (g_2 a_{k_3}^\dagger a_{k_4} + g_3 b_{k_3}^\dagger b_{k_4}) \,.
\tag{24}
$$

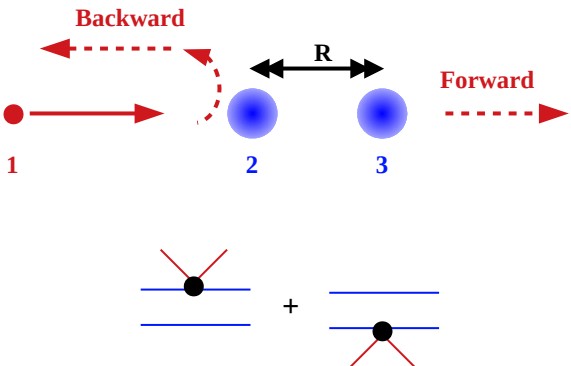

Figure 6: Forward and backward scattering of a scalar particle 1 off the bound state formed by distinguishable particles 2 and 3, also scalar, and the two connected scattering diagrams in Born approximation to the Lagrangian in Eq. (21).

To proceed, we need to specify the scattering kinematics. The momentum transfer from the projectile is $\Delta := |\mathbf{q}_1 - \mathbf{q}_1'|$. If the total mass of the 23 bound state is taken to be $M$, the target recoils with velocity $v = \frac{\Delta}{\sqrt{M^2 + \Delta^2}}$. This takes it to a boosted state that can be approximated by

$$\langle \psi_{23} | K_v^\dagger = \int \frac{d^3 q_2'}{(2\pi)^{3/2}} \frac{d^3 q_3'}{(2\pi)^{3/2}} \sqrt{\frac{a^6}{\pi^3}} e^{-\frac{a^2}{2}(\mathbf{q}_2'^2 + \mathbf{q}_3'^2)} \langle 0 | b_{q_3^B} a_{q_2^B} e^{+\frac{iR}{2}(q_3' - q_2')_z} \tag{25}$$

where the boosted momenta are given by the usual Lorentz transformation, for example, that of particle 2,

$$q_2^B = \frac{1}{M} \left( E_2' \Delta + q_2' \sqrt{M^2 + \Delta^2} \right). \tag{26}$$

A change of variables from $q_i^B$ to $q_i'$ takes the boost off the field quanta to expose it in the wavefunction (which is then Lorentz contracted).

We can now collect all the pieces to mount the scattering amplitude in first order perturbation theory, yielding the scattering of particle 1 (in an eigenstate of momentum) off the 23 bound state (that is pushed to a boosted frame),

$$\mathcal{M} \propto \left( \langle \mathbf{q}_1' | \otimes \langle \psi_{23} | K_v^\dagger \right) V \left( | \psi_{23} \rangle \otimes | \mathbf{q}_1 \rangle \right)$$
$$= \int \frac{d^3 q_2^B}{(2\pi)^3} \frac{d^3 q_3^B}{(2\pi)^3} \frac{d^3 q_2}{(2\pi)^3} \frac{d^3 q_3}{(2\pi)^3} 4 \frac{a^6}{\pi^3} e^{-\frac{a^2}{2}(q_2^2 + q_3^2 - q_2'^2 - q_3'^2)} e^{-i\mathbf{X}_3 \cdot (\mathbf{q}_3 - \mathbf{q}_2)} e^{+i\mathbf{X}_3 \cdot (\mathbf{q}_3' - \mathbf{q}_2')} \cdot (2\pi)^6$$
$$\times \Big[ g_2 \, \delta^{(3)}(\mathbf{q}_3' - \mathbf{q}_3) \, \delta^{(3)}((\mathbf{q}_1 - \mathbf{q}_1') - (\mathbf{q}_2' - \mathbf{q}_2))$$
$$+ g_3 \, \delta^{(3)}(\mathbf{q}_2' - \mathbf{q}_2) \, \delta^{(3)}((\mathbf{q}_1 - \mathbf{q}_1') - (\mathbf{q}_3' - \mathbf{q}_3)) \Big] . \tag{27}$$

Evaluating the phase factor $\Phi$ in the third line of the integrand with the momentum conservation delta distributions yields

$$\Phi = g_2 \, e^{-i\mathbf{X}_3 \cdot (\mathbf{q}_3 - \mathbf{q}_3')} \Big|_{\mathbf{q}_3 = \mathbf{q}_3'} e^{-i\mathbf{X}_3 \cdot (\mathbf{q}_2' - \mathbf{q}_2)} \Big|_{\mathbf{q}_2' - \mathbf{q}_2 = \mathbf{q}_1 - \mathbf{q}_1'}$$
$$+ g_3 \, e^{-i\mathbf{X}_3 \cdot (\mathbf{q}_2' - \mathbf{q}_2)} \Big|_{\mathbf{q}_2 = \mathbf{q}_2'} e^{-i\mathbf{X}_3 \cdot (\mathbf{q}_3 - \mathbf{q}_3')} \Big|_{\mathbf{q}_3' - \mathbf{q}_3 = \mathbf{q}_1 - \mathbf{q}_1'} , \tag{28}$$

so that two of the exponentials become simply unity and two become evaluated in the external momenta of the scattered particle 1, so they factor out of the integration.

This multiplicative complex factor (the rest of the matrix element in Eq. (27) is purely real) is then

$$\Phi = g_2 e^{-i\mathbf{X}_3 \cdot (\mathbf{q}_1 - \mathbf{q}_1')} + g_3 e^{+i\mathbf{X}_3 \cdot (\mathbf{q}_1 - \mathbf{q}_1')} . \tag{29}$$

The phases can be interpreted as the diffraction of the quantum beam by the finite sized target. In the collinear kinematics of figure 6, this is

$$\Phi = g_2 e^{-i\frac{R}{2}\Delta} + g_3 e^{+i\frac{R}{2}\Delta} \tag{30}$$

or, more generally, noting that $\Delta_z = \sqrt{|t|}$ and with $t = -\frac{s-4m^2}{2}(1-x)$,

$$\Phi = g_2 e^{-i\frac{R\sqrt{s-4m^2}}{2}\sqrt{\frac{1-x}{2}}} + g_3 e^{+i\frac{R\sqrt{s-4m^2}}{2}\sqrt{\frac{1-x}{2}}} . \tag{31}$$

The factor with $g_2$ corresponds to scattering by the leading particle that the beam finds before the center of mass, and is analogous to the factor that we found earlier in subsection 1.1 for a spherical shell.

When extending $s$ to the complex plane, $s \to s_R + i s_I$, it yields an exploding contribution $\sim e^{+s_I}$ that makes the closing of a great circle over the upper half plane $\mathbb{C}_+$ unfeasible. Thus, dispersion relations in the complex $s$-plane are not generically granted except for fixed $t$, including forward $t = 0 = \theta$ scattering; in other cases, such as when attempting a dispersive analysis of partial waves, one needs to multiply the scattering amplitude by a small enough compensating exponential factor and write a dispersion relation for the modified amplitude. This procedure seems to introduce a theoretical uncertainty, associated to the soft physics of the target, that is very difficult to reduce.

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
