# Peer review of "Smearing of causality by compositeness divides dispersive approaches into exact ones and precision-limited ones"

_SciPost Physics Core, doi:SciPost Phys. Core 5, 016 (2022)_

## Round 1 · Referee Report · Anonymous (Referee 1) · 2020-11-18

Strengths

  • Pedagogic introduction
  • Elegant tool (dispersion relations)
  • Relevant for phenomenology

Weaknesses

  • Given existence of valid fixed-t dispersion relations, it is not so clear where the problem is

Report

Dear Editor,

the authors discuss dispersion relations that originate from causality and unitarity, and relate the IR and UV parts of the scattering amplitude. They point out a potential problem in fixed-angle dispersion relations; in particular they show that the scattering off objects of finite size introduces an apparent violation of causality, despite the microscopic physics being causal.

The topic is in principle interesting, as dispersion relations are an important tool in particole physics. However, as the authors point out, this problem doesn’t appear in fixed-t or forward dispersion relations, which are typically employed in particle physics. So, I believe the authors should make more effort to motivate why plain dispersion relations for partial waves are a priori better than fixed-t dispersion relations (and show that they contain more information).
Also, it is not clear to me how a computation on the righthand-cut-only suffices.
So, rather than pointing out a specific situation in which a specific dispersion relation doesn’t work, I’d see more useful an article that shows what instead works. The authors have some ideas in the 7th paragraph of the conclusions, that should be investigated more.

Requested changes

1) Motivate why plain dispersion relations are important

2) What does the violation of causality imply for the analyticity properties of partial waves?

3) Explain why the authors think the computation on the righthand cut is sufficient. (see p.7)

4) What do the authors mean when they say on p.7 “the important case of QFT”; what other case do they have in mind?

5) The link with LHC physics is a bit poor: can the authors explain better how this would impact the search for SM devviations?

6) extend discussion on polynomials and t and t’ in the conclusions.

  • validity: good
  • significance: ok
  • originality: good
  • clarity: good
  • formatting: good
  • grammar: excellent

Author:  Felipe J. Llanes-Estrada  on 2020-11-24  [id 1063]

(in reply to Report 1 on 2020-11-18)
Category:
answer to question
reply to objection
pointer to related literature

The referee raises some good points that we quickly address.

We should first like to recall that fixed-t dispersion relations were indeed safely constructed in the 50s, partly to circumvent this type of problems. But after six or seven decades, brief mentions in classic articles such as our reference [10] or nonrelativistic discussions in books long out of print such as our reference [7] have largely disappeared from the collective memory, so that practitioners sometimes ignore the distinction between those and other types of dispersion relations. Because partial-wave amplitudes are convenient to analyse new-physics or also hadron resonances (keeping in mind that they are sensitive to only one angular momentum channel at a time, isolating new particles), they remain in use. They satisfy good analyticity properties leading the community to expect them to also automatically satisfy good dispersion relations. But as we show, this is not automatic because of divergent behavior deep in the complex plane and care should be exercised (by deploying the dispersion relation for the Inverse Amplitude, for example) .
We do not believe that partial-wave nor fixed-t dispersion relations are better than the other: they are useful for different problems.

Respect to the comment about the impact of the left-hand cut, a short answer could be that, because physics happens at s>threshold on the right cut, and Cauchy's theorem suppresses contributions far in the complex plane by a 1/(s-s') denominator, the integral over the right cut is numerically more important for physical applications. Of course, a more detailed analysis is called for, and a follow-up preprint by a different author collaboration https://inspirehep.net/literature/1826204 addresses the systematic theory uncertainties in the Inverse Amplitude Method, prominently featuring the left-cut. Interested readers can delve into that work for detail.
That method does work and has been discussed at length in the literature (including that recent preprint and the many references therein). Plain PW dispersion relations can also work in restricted kinematic domains where they can be derived from fixed-t dispersion relations, but the community should be continuously aware that restrictions exist and they cannot be blindly applied (and the same for fixed-angle dispersion relations or others).

The referee is of course aware that S-matrix theory is more general than local quantum field theory applied to physical kinematics. Non pointlike interactions or objects (e.g. strings) can be cast in the S-matrix formalism and predict scattering amplitudes with the required properties. That was a passing comment and we have no particular interest in pursuing the matter.

The LHC link however is of interest for our phenomenological physics program: if/when separations from the SM are found, they will likely be cast in the form of an Effective Theory for the SM particle content. Dispersion relations can then play a role in extending the reach of those low-energy measurements to ascertain what may await at higher energies.

We thank all parties for the collegial consideration of our manuscript, and particularly the reviewer for his or her dedication and appraisal of the article.

Anonymous on 2021-06-03  [id 1484]

(in reply to Felipe J. Llanes-Estrada on 2020-11-24 [id 1063])

Dear colleague, in preparation for a resubmission of the manuscript, here is the list of changes implemented.

1) Motivate why plain dispersion relations are important

RESPONSE: we have commented about this topic in the introduction, though not extensively, as we believe this article needs to remain of moderate size consistently with the amount of material reported.

2) What does the violation of causality imply for the analyticity properties of partial waves?

RESPONSE: This can be read off equation (4). The lower integration limit needs to be set to -2R/c sin(theta/2) < 0 since the time-dependent amplitude only vanishes for earlier times. This change does not affect the convergence at tau-> infinity, so that the resulting t_J(E) is still analytic in the upper-energy plane; but it grows (exponentially) for increasingly positive Im(E). That is, a finite target yields an analytic function but a nonconvergent contour integral. Only when the signal can be arbitrarily advanced, and tau-> -infinity needs to be taken in the time integration, is analyticity completely lost. This comment has now been added to subsection 1.1.

3) Explain why the authors think the computation on the righthand cut is sufficient. (see p.7) RESPONSE: we do not think it is sufficient for a full evaluation of the amplitude, and we discuss it at length in the concurrent extensive paper submitted to SciPost Physics, issued as e-Print: 2010.13709 [hep-ph]. There is minimum explanation here after Eq.(13). Remember that we are only discussing the uncertainties, and those due to the two cuts should be linearly summed, being conservative.

4) What do the authors mean when they say on p.7 “the important case of QFT”; what other case do they have in mind? RESPONSE: S-matrix theory is more general and can be applied to string theory, to nonrelativistic quantum mechanics and other approaches; this is now explicitly stated at the end of section 2.

5) The link with LHC physics is a bit poor: can the authors explain better how this would impact the search for SM deviations? RESPONSE: We have tried to show it by reorganizing section 2 into subsections and adding minor paragraphs elsewhere as needed.

6) extend discussion on polynomials and t and t’ in the conclusions. RESPONSE: we have added this suggested comment, known from the usage of the Omnès- Mushkelishvili method.

---

## Round 1 · Referee Report · Anonymous (Referee 2) · 2020-11-29

Strengths

1- Rises an apparent problem 2- Authors are experts in this field 3- Paper is well written 4- Issue being discussed that appears to be of potential relevance

Weaknesses

1- At the end of the day, there is no real problem. Useful or confusing? 2- Authors mix range of interaction with scale of compositeness 3- Maybe misleading to the reader

Report

This manuscript has brought more questions than answers to this referee. I agree with the general discussion in Sect 1, although in my opinion the notation should be improved (see below). However I find the discussion presented in Sect 2 rather confusing and misleading.

In effective theory treatments, pions (or longitudinal W for that matter) are treated as point-like objects. Period. Then, none of the points discussed in Sect 1 would immediately apply. We of course know that pions are not point-like or elementary, but Ws could still be to the best of our knowledge. Of course they have a range of interactions (we may take that as 1/M_W in the case of the electroweak bosons, for instance, but this is in principle unrelated to the scale of compositeness. A similar comment could apply to pion scattering.

Being treated as point-like objects I cannot see the relevance of the discussion displayed in Sect. 1

Sect. 3 is fine, but I do not see much new there.

Requested changes

  1. In Sect 1 I would recommend replacing f(x) for a physical amplitude to make the reader understand what we are talking about

2- Authors should propose an example where the connection to the problem presented in Sect. 1 is obvious (not the case now).

3- Some English checks needed: there is a "safe for" on p4 that makes no sense. cm as center-of-mass is not defined.

  • validity: low
  • significance: ok
  • originality: good
  • clarity: good
  • formatting: good
  • grammar: good

Author:  Felipe J. Llanes-Estrada  on 2021-06-03  [id 1485]

(in reply to Report 2 on 2020-11-29)

We also thank the second reviewer for his careful reading of the manuscript and suggestions, and we believe that we can address his/her criticism.

REFEREE: In effective theory treatments, pions (or longitudinal W for that matter) are treated as point-like objects. Period. Then, none of the points discussed in Sect 1 would immediately apply. We of course know that pions are not point-like or elementary, but Ws could still be to the best of our knowledge.

RESPONSE: The referee is certainly right that for many purposes one can treat Goldstone bosons as pointlike objects. As he states, this is an approximation for pions, and might or might not be correct for W_L. Our point is that, while (RxE) (size x energy) may be small at an accelerator where the energy might be insufficient to probe the object's size, when closing the Cauchy contour in the complex plane, the exponent (Rx Im(E) ) becomes arbitrarily large. We believe that we can restate our confidence in the results of section 1, they are correct in the presence of an underlying structure, even when that structure is not yet probed at a reached energy.

REFEREE: Of course they have a range of interactions (we may take that as 1/M_W in the case of the electroweak bosons, for instance, but this is in principle unrelated to the scale of compositeness. A similar comment could apply to pion scattering. RESPONSE: That statement is correct. The scale of compositeness and the range of interaction can be unrelated, if the mass of the force carrier is very dissimilar to the mass of the source (e.g. the pion and the nucleon). We have improved the logic by clarifying that our considerations apply to a) finite range interactions b) apparently zero range/delta function interactions when the object is composite at any scale.

REFEREE: Being treated as point-like objects I cannot see the relevance of the discussion in Sect. 1 RESPONSE: A dispersion relation is sensitive to very large energies that probe the underlying scale. One can escape this fact over the real axis because the exponential phase factor is 1, so that the divergence can typically be tamed by subtractions (providing an s^-n denominator). But not with an imaginary part of E.

Sect. 3 is fine, but I do not see much new there. RESPONSE: Certainly. Sec. 3 is provided for readers of various backgrounds to clearly apprehend that the type of dispersion relations there discussed are safe; that is, to show with some detail which dispersion relation demonstrations are, and which ones are not (and why) affected by large Im(E) divergences. We think that it is important to keep these comments to avoid confusion by off-field readers that may think that we cast doubt on the entire dispersion-relation approach, but we will agree to drop them if deemed necessary.

Requested changes 1. In Sect 1 I would recommend replacing f(x) for a physical amplitude to make the reader understand what we are talking about. RESPONSE: we have changed it for a partial wave t_J starting in Eq. (2); some rewriting of the text was correspondingly necessary and has been carried out.

2- Authors should propose an example where the connection to the problem presented in Sect. 1 is obvious (not the case now). RESPONSE: Any scattering process in 2 or greater dimension, where one or both objects are of finite size, or where the interaction is finite range, will suffer from that exponential behaviour in the complex plane and care with the formulation of dispersion relations will be needed. Examples include: photon-hadron scattering, hadron-hadron scattering, photon-atom scattering, and so on. This comment has now been incorporated into the manuscript.

3- Some English checks needed: there is a "safe for" on p4 that makes no sense. cm as center-of-mass is not defined.

RESPONSE: We have proofread the manuscript and improved as needed. We understand that the expression "safe for" or "save for" may be unfamiliar to some readers (it was imported from french into english, in our understanding, but seems to be correct usage) and have substituted it by the more common "except for". We now have defined "center of mass" explicitly, though this is almost universal in the theoretical physics literature.

---

## Round 2 · Referee Report · Anonymous (Referee 2) · 2021-9-19

Report
a complexified energy variable is not granted.
As a shortcoming I would mention that, having blown the whistle, they do not present a convincing calculation or example showing the actual and detailed relevance of the effect. Instead they introduce a modified partial wave (without obvious physical meaning, as they state) but having a good behaviour in the upper half plane. In the integrals considered in Eq. 11 no complex contour is required and indeed they work with a linearized version of the exponential that yields a good behaviour. So, altogether it is not clear to the reader the relevance of the whole study. The only clear message is that there may be a problem.
Two last comments: In the introduction it is said that single variable dispersion relations are unafected in any case. Yet the authors explain in some detail the dispersion relation involved in g-2. I do not see the necessity of this digression. Second, section 2.2 on W_L scattering is probably unnecessary as there is no evidence whatsoever that they composite.

Anonymous on 2021-09-18 [id 1765]
The manuscript in its present form shows a clear improvement with respect the version thta was previously reviewed. The authors have taken into account the criticisms and comments in my previous reports. As it stands the paper sends a word of caution when using multi-variable dispersion relations because the convergence in the upper half plane in integrals over
a complexified energy variable is not granted.
As a shortcoming I would mention that, having blown the whistle, they do not present a convincing calculation or example showing the actual and detailed relevance of the effect. Instead they introduce a modified partial wave (without obvious physical meaning, as they state) but having a good behaviour in the upper half plane. In the integrals considered in Eq. 11 no complex contour is required and indeed they work with a linearized version of the exponential that yields a good behaviour. So, altogether it is not clear to the reader the relevance of the whole study. The only clear message is that there may be a problem.
Two last comments: In the introduction it is said that single variable dispersion relations are unafected in any case. Yet the authors explain in some detail the dispersion relation involved in g-2. I do not see the necessity of this digression. Second, section 2.2 on W_L scattering is probably unnecessary as there is no evidence whatsoever that they composite.

---

## Round 3 · Author Response

Dear Editor,
Thank you for forwarding the last rounds of referee comments. We regret not having responded earlier, it has been quite
a hectic trimester with the overburden of teaching under pandemic conditions and there was a lot of backlog to clear.
We should like to respond to these last observations as follows.

REFEREE: The manuscript in its present form shows a clear improvement with respect the version thta was previously reviewed. The authors have taken into account the criticisms and comments in my previous reports. As it stands the paper sends a word of caution when using multi-variable dispersion relations because the convergence in the upper half plane in integrals over
a complexified energy variable is not granted.

RESPONSE: We thank the referee for the appreciative comment.

REFEREE: As a shortcoming I would mention that, having blown the whistle, they do not present a convincing calculation or example showing the actual and detailed relevance of the effect. Instead they introduce a modified partial wave (without obvious physical meaning, as they state) but having a good behaviour in the upper half plane.
In the integrals considered in Eq. 11 no complex contour is required and indeed they work with a linearized version of the exponential that yields a good behaviour. So, altogether it is not clear to the reader the relevance of the whole study. The only clear message is that there may be a problem.

RESPONSE: We can see the referee's point. Indeed, because the modified partial wave has good convergence properties,
the contour at infinity is well-behaved and the integration along the real-axis only is granted.
This allows to estimate what separation from the actual dispersion relation for the physical partial wave
can be expected in the absence of other theoretical support, which can be found for certain
kinematic configurations in fixed-t dispersion relations, but not generally for all kinematics. Auxiliary
functions such as the one introduced allow to obtain partial-wave or fixed-angle dispersion relations which
can then be related to the physical scattering functions. The linearized version is offered for clarity, but
there is no obstacle to use the full exponential (numerically) as is the case in Eq.(13).
We have attempted to make these points clearer in the manuscript. For this, section 2 has been reorganized,
with a break just before Eq.(8) to consolidate this discussion at the beginning of a subsection,
and a paragraph has been added there.

REFEREE: Two last comments: In the introduction it is said that single variable dispersion relations are unafected in any case. Yet the authors explain in some detail the dispersion relation involved in g-2. I do not see the necessity of this digression.

RESPONSE: The community is immersed in reviewing every aspect of the g-2 computation as well as the experimental setup,
in view of the disagreement with the standard model recently sharpened by the FNAL g-2 data. In view of this effort,
it appears to us of nonnegligible interest to very quickly review the foundation of the corresponding dispersion
relation, explicitly showing that it is not a problematic one, since a lot of the theory effort hangs from it.

REFEREE: Second, section 2.2 on W_L scattering is probably unnecessary as there is no evidence whatsoever that they composite.

RESPONSE: Indeed, there is no current evidence of their compositeness, but there is ample work preparing for the eventuality
that such discovery would be made. We have added reference to a recent review by Dobado and Espriu on the topic.
In addition, when/if WLWL scattering data becomes available, it can be tested against a dispersion relation to
ascertain its possible composite or pointlike nature. The discussion has been slightly revised.
We think this is an important potential application of dispersion relations in high-energy physics that should not
be discounted.

We have reread the manuscript and made minor text improvements wherever it seemed appropriate.
With these changes, we hope the editorial now finds it acceptable.

---

## Editorial Decision

published